# An Understanding of How GDP, Unemployment and Inflation Interact and Change across Time and Frequency

Yegnanew A. Shiferaw 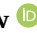

Department of Statistics, University of Johannesburg, Auckland Park, Johannesburg 2006, South Africa; yegnanews@uj.ac.za; Tel.: +27-011-559-3024

**Abstract:** The main aim of this paper is to examine the dynamic relationship between the three pillars of the economy: unemployment, inflation, and GDP in Ethiopia using the cross-wavelet transform (XWT) analysis, the multivariate Student-*t* generalized autoregressive score (GAS) model, and the autoregressive distributed lag (ARDL) model. The dynamics between the three indicators were also investigated using the Toda–Yamamoto (TY) causality test. The empirical findings from the XWT method suggest a relationship between unemployment, inflation, and GDP, though the relationship varies over time and frequency. The estimation results from the multivariate Student-*t* GAS model show that the correlation between unemployment (overall, male, female, and youth) and inflation is highly significant, indicating that the correlation is dynamic. A dynamic relationship exists between GDP and unemployment, except for females and young people. The ARDL approach's findings showed that unemployment significantly negatively impacted GDP. However, it was found that inflation significantly increased GDP. The general conclusion drawn from this study's findings is that unemployment significantly affects GDP and inflation. Therefore, the government should aggressively implement policies to reduce unemployment, especially youth unemployment. Additionally, the administration must rehabilitate the country's badly damaged economy and formalize a lasting cessation of hostilities between the federal government and the Tigray People's Liberation Front (TPLF).

**Keywords:** Gross Domestic Product; generalized autoregressive score model; inflation; time varying correlation; unemployment; cross-wavelet transform

## 1. Introduction

Ethiopia has one of the most dynamic economies in Sub-Saharan Africa (Dorosh and Mellor 2013), as well as one of the largest populations, water resource potential (Hailu 2022), hydropower potential to generate hydropower with a capacity of 45,000 Mega Watt (MW) (International Energy Agency 2019), and a total surface area of more than 1.1 million square kilometers (Khatami et al. 2020). Ethiopia's economic growth rates have been the subject of several studies. For instance, the country has seen impressive growth and poverty reduction from 2010/11 to 2014/15, with GDP growth averaging 10.1% or approximately 8% per capita growth (The Federal Democratic Republic of Ethiopia 2016). Poverty has been significantly reduced, with a low Gini index of 30% by international and Sub-Saharan African standards (The Federal Democratic Republic of Ethiopia 2016). The nation's GDP has grown by double digits, the second-fastest in Africa after Angola (Deloitte 2014; Rodrik 2018). The country's growth miracle assisted the Ethiopian government in (i) gaining the interest of various foreign firms (Deloitte 2014), and (ii) lowering rural poverty by increasing economic productivity as a whole (Rodrik 2018).

Despite consecutive GDP growth, Ethiopia's macroeconomic performance deteriorated in 2015–2016 due to a severe drought and an unfavorable global climate. As a result, output grew slower in 2015–2016 (The Federal Democratic Republic of Ethiopia 2016). Aside from severe drought and a deteriorating global climate, high unemployment has become

common in Ethiopian and African cities in a broad sense (Daniel 2011; Mokona et al. 2020). For instance, youth unemployment rose by 60% between 1990 and 1994, and youngsters with secondary education experienced the highest unemployment rates (Krishnan 1996). Urban unemployment was 25% in 2015 and 25.3% in 2018, which is significantly higher among the youth and is primarily a social and health concern, notwithstanding minor gains in net employment prospects since 2003 (Mokona et al. 2020; Shiferaw 2017). Furthermore, Ethiopia had the highest rate of month-to-month food inflation among developing countries between 2008 and 2011, at 3.5% per month, owing primarily to rising food prices for maize, wheat, and teff (Bachewe and Headey 2017). Following the 15% devaluation of the Ethiopian currency (the birr) in October 2017, there has been rising pressure on inflation recently. In April 2018, the annual general price level climbed by 13.7%, food prices increased by 16.1%, and non-food item prices increased by 10.8% (Quarterly Economic Brief 2018).

The world has been experiencing an unprecedented situation due to the coronavirus disease (COVID-19) since the beginning of 2020. COVID-19 has been a massive health and economic disaster for the world (Hensher 2020; Shiferaw 2021). Ethiopia, Africa's second-most populous country and one with a rapidly expanding economy before the outbreak, was hit hard by the pandemic, as were many other African countries. Furthermore, many countries saw unexpected unemployment increases due to this pandemic (Hensher 2020). With 345 million full-time equivalent jobs lost globally in the third quarter of 2020 alone, unemployment was one of the world's significant challenges (ILO 2020). The COVID-19 pandemic has impacted employment in Ethiopia and many other countries, leaving many households unemployed.

On 5 November 2020, the Tigray People's Liberation Front (TPLF) and the Ethiopian federal government engaged in armed conflict, but difficulties persisted. The TPLF was a major party in the Ethiopian People's Revolutionary Democratic Front (EPRDF), a four-party coalition that dominated the country for nearly three decades (Meester et al. 2022). This conflict led to a civil war that ravaged large portions of the nation and killed countless people. Due to the civil war, the country had severe political, racial, and economic problems that led to substantial job losses and the highest inflation in the nation's history (Matshanda 2022).

For several reasons, understanding the relationships between unemployment, inflation, and GDP is essential for a nation still fighting to transcend poverty. For instance, a successful transition to a manufacturing and service-oriented economy depends on having a better grasp of the characteristics of the urban labor market (Vera-Toscanoa et al. 2020). The leading indicators that the public and policymakers pay special attention to and check include unemployment, inflation, and GDP (Adams et al. 2021; Debaere 2008). They serve as an economy's scorecard and provide an impression of its general health (Debaere 2008). This study is novel because it fills a critical knowledge gap about the relationship between GDP changes, unemployment rates, and inflation in Ethiopia, the biggest landlocked country in the world and the second-most populous country in Africa. According to the author, more research is needed to determine how Ethiopia's GDP, unemployment, and inflation are related. In order to bridge this gap, this study looks at the dynamic relationships between these metrics in Ethiopia. As another way of putting it, this study empirically investigates and answers the following question: How do the three indicators interact over time and frequency? The solution to this problem will provide new insight into the three indicators and help Ethiopian policymakers with sectoral resource allocation in general and policies in particular.

Various statistical approaches have been used to investigate the empirical relationships between unemployment, GDP, and inflation. These include time series analysis, geographic panel-data models, panel regression, Granger causality tests, and regression analysis (Abdullah et al. 2020; Batrancea 2021a, 2021b; Batrancea et al. 2022; Bein and Ciftcioglu 2017; Phillips 1958; Schubert and Kroll 2016; Simona et al. 2019). Many of these approaches relied on linear and stationary assumptions to characterize time series relationships and

determine the temporal-lag time series changes (Yu and Lin 2015). However, due to the effects of socioeconomic factors such as political instability (Matshanda 2022), climate change (Wendimu 2021), the COVID-19 pandemic (Hensher 2020; ILO 2020), and so on, the interactions between unemployment, GDP, and inflation in Ethiopia can frequently vary over time and present non-stationary relationships. In this paper, we apply the cross-wavelet transform (XWT) method to study the relationship between unemployment, GDP, and inflation in Ethiopia. The XWT identifies links between non-stationary time series by measuring the correlation between two time series in the time-frequency domain (Banerjee and Mitra 2014; Dey et al. 2010; Grinsted et al. 2004). Furthermore, to examine the time-varying correlations between the three indicators, this paper employs a multivariate Student-*t* Generalized Autoregressive Score (GAS) model (Creal et al. 2013; Harvey 2013). This model provides policymakers with new analytical tools for capturing the dynamic characteristics between the three indicators (Jiang et al. 2022). The study also used the autoregressive distributed lag (ARDL) approach, a well-known method developed by (Pesaran and Shin 1999; Pesaran et al. 2001), to examine the relationship between the three indicators. According to (Duasa 2007), the ARDL method is suitable for simultaneously generating short-run and long-run elasticities for a small sample size.

In conclusion, the main objective of the paper was to investigate the dynamic interactions between the three leading indicators (GDP, unemployment, and inflation) using time series analysis and econometric methods in the time domain, such as the multivariate Student-t GAS model and the ARDL model, and time series in the frequency domain, such as the XWT method, using data spanning three decades.

The remaining sections of the paper are structured as follows: Section 2 describes the methods used. Empirical findings are presented in Section 3. In Section 4, there is a discussion. The research comes to an end in the last segment.

## 2. Data and Empirical Methods

### 2.1. Empirical Methods

#### 2.1.1. The Cross Wavelet Transform

The wavelet analysis can be applied using discrete or continuous wavelet transform approaches. The continuous wavelet transform has an advantage over the discrete wavelet transform in that it allows for the selection of wavelets based on the duration of the data. The dynamic relationships between inflation, GDP, and unemployment are examined in this research using a wavelet approach over a range of time horizons. This method offers insightful information on how the time series behaves in the frequency and time domains.

The XWT is a measure of similarity between two waveforms (Banerjee and Mitra 2014), revealing information about the phase relationship and regions with high common power. In addition, an XWT measures the correlation between two time series in the time-frequency domain, so it is more appropriate for finding existing links between two-time series. The XWT of two-time series $x_t$ and $y_t$ is defined as:

$$W_{xy}(\tau, s) = W_y(\tau, s) W *_x (\tau, s) \tag{1}$$

where $W_y(\tau, s)$ and $W *_x (\tau, s)$ are the wavelet transform of $y_t$ and $x_t$, respectively; and $W *_x$ represents the complex conjugate of $W_x$. Furthermore, $W_{xy}(\tau, s)$ argument represents the relative phase between two-time series at a local level (Grinsted et al. 2004). In the XWT plots, warmer colors have strong correlations, but colder colors exhibit lesser correlations and interrelations. Additionally, (i) the red regions denote the most important relationship (high-intensity levels), indicating that extreme movements (Firouzi and Wang 2019; Hathroubi and Aloui 2016; Le 2019); and (ii) the blue regions denote the lowest relationship (low-intensity levels) (Firouzi and Wang 2019; Hathroubi and Aloui 2016; Le 2019).

### 2.1.2. The Multivariate Student-*t* GAS Model

Harvey (2013) and Creal et al. (2013) proposed a new type of observation-driven model for modeling time variation in parametric models based on the score function, known as the dynamic conditional score (DCS) model and the generalized additive score (GAS) model, respectively.

Suppose the observation vector $\mathbf{y}_t \in \Re^k$ follows a standardized Student-*t* distribution with $\nu$ degrees of freedom. The observation density of $y_t$ is given by

$$p(y_t|\Sigma_t : \nu) = \frac{\Gamma((\nu + k)/2)}{\Gamma(\nu/2)[(\nu - 2)\pi]^{k/2}|\Sigma_t|^{1/2}} \times \left[1 + \frac{y_t'\Sigma_t^{-1}y_t}{(\nu - 2)}\right]^{\frac{-(\nu+k)}{2}}, \tag{2}$$

where $\Sigma_t$ is the covariance matrix of $y_t$.

The time-varying parameters for a multivariate Student-*t* distribution can be collected in the vector $f_i$ and specify the autoregressive updating function by

$$f_{t+1} = \omega + \sum_{i=1}^{p} A_i s_{t-i+1} + \sum_{j=1}^{q} B_j f_{t-j+1}, \tag{3}$$

where $\omega$ is a vector of constants, $A_i$ and $B_j$ are coefficient matrices with dimensions $i = 1, \cdots, p$ and $j = 1, \cdots, q$, $s_t$ is a scaled function of current and past data.

Following Engle (2002), the variance-covariance matrix can be decomposed for the multivariate GAS model as follows:

$$\Sigma_t = \mathbf{D}_t \mathbf{R}_t \mathbf{D}_t, \tag{4}$$

where $\mathbf{D}_t$ is the diagonal time-varying standard deviation matrix and $\mathbf{R}_t$ is the correlation matrix containing conditional correlations. For the multivariate Student-*t* density with time-varying $\mathbf{D}_t$ as well as $\mathbf{R}_t$, the correlation matrix $\mathbf{R}_t$ can be decomposed as follows:

$$\mathbf{R}_t = \mathbf{\Delta}_t^{-1} \mathbf{Q}_t \mathbf{\Delta}_t^{-1}, \tag{5}$$

where $\mathbf{Q}_t$ is a symmetric positive definite matrix and $\mathbf{\Delta}_t^{-1}$ is a diagonal matrix with whose non zero elements equal the square root of the diagonal elements of $\mathbf{Q}_t$.

### 2.1.3. ARDL Approach

In addition to the wavelet squared coherence and GAS model, this paper used the ARDL approach to examine the relationship between GDP, unemployment, and inflation. The ARDL model proposed by (Pesaran and Shin 1999; Pesaran et al. 2001) has several advantages. For example, in the ARDL model, the past and current values of explanatory variables influence the response variable. Moreover, the response variable is also explained by its past values. As suggested by (Batrancea et al. 2022), we used the GDP growth rate as a proxy for the rate of economic growth (i.e., a dependent variable) and variables like inflation and unemployment as independent variables. In this study, we estimate the following ARDL models:

$$\text{GDP}_t = c + \sum_{i=1}^{m} \alpha_i \text{GDP}_{t-i} + \sum_{i=0}^{p} \beta_i \text{unemp}_{t-i} + \sum_{i=0}^{q} \gamma_i \text{inflation}_{t-i} + \epsilon_t \tag{6}$$

where $\epsilon_t$ is a white noise error term, $m, p, q$ are the lagged values of the response and explanatory variables, respectively. The autocorrelation in the estimation residuals can be tested using the Durbin-Watson (DW) (Durbin and Watson 1950) test. This standard test does not rely on a large sample size approximation. We selected the best candidate models using the Akaike Information Criterion (AIC) and the log-likelihood (Loglik) function. Models with higher Loglik values and lower AIC values are preferred.

*2.2. Data Description*

This study uses the yearly unemployment rates (total, youth, female, and male), inflation, and GDP from 1991 through 2021, taken from the Federal Reserve Bank of St. Louis. Table 1 reports the descriptive statistics of these indicators. All of these indicators have a positive mean and median. These indicators generally have low standard deviations except for GDP and inflation, with the highest standard deviations of 30.71 and 11.06. All of the indicators in this paper have positive skewness. Additionally, all indicators have positive kurtosis, indicating that the conditional distribution has a heavy tail.

**Table 1.** Descriptive statistics of the raw data of unemployment, GDP, and inflation.

|  | Min (%) | Max (%) | Mean (%) | Median (%) | St. Dev. (%) | Skewness | Kurtosis |
|---|---|---|---|---|---|---|---|
| Total unemployment | 2.25 | 3.71 | 2.82 | 2.69 | 0.51 | 0.32 | −1.51 |
| Youth unemployment | 3.43 | 5.72 | 4.22 | 3.88 | 0.73 | 0.38 | −1.44 |
| Female unemployment | 2.94 | 4.99 | 3.55 | 3.27 | 0.64 | 0.86 | −0.75 |
| Male unemployment | 1.65 | 2.97 | 2.20 | 1.96 | 0.52 | 0.30 | −1.74 |
| Inflation | −8.24 | 44.37 | 11.85 | 9.99 | 11.06 | 0.74 | 0.80 |
| GDP | 7.51 | 99.27 | 32.66 | 15.31 | 30.71 | 0.97 | −0.59 |

## 3. Results

This section presents the results of the wavelet analysis and the multivariate GAS model. The lead-lag connectivity between unemployment, GDP, and inflation was examined using XWT techniques. The heat map of a visual correlation matrix across unemployment (overall, youth, female, and male), GDP, and inflation is shown in Figure 1. The intensity of the colored boxes in the "shaded boxes" indicates the magnitude of the correlation, with red denoting a positive correlation and blue representing a negative correlation. The map shows a negative correlation between unemployment and inflation rates, unemployment rates, and GDP. It is worth noting that GDP has a positive relationship with inflation.

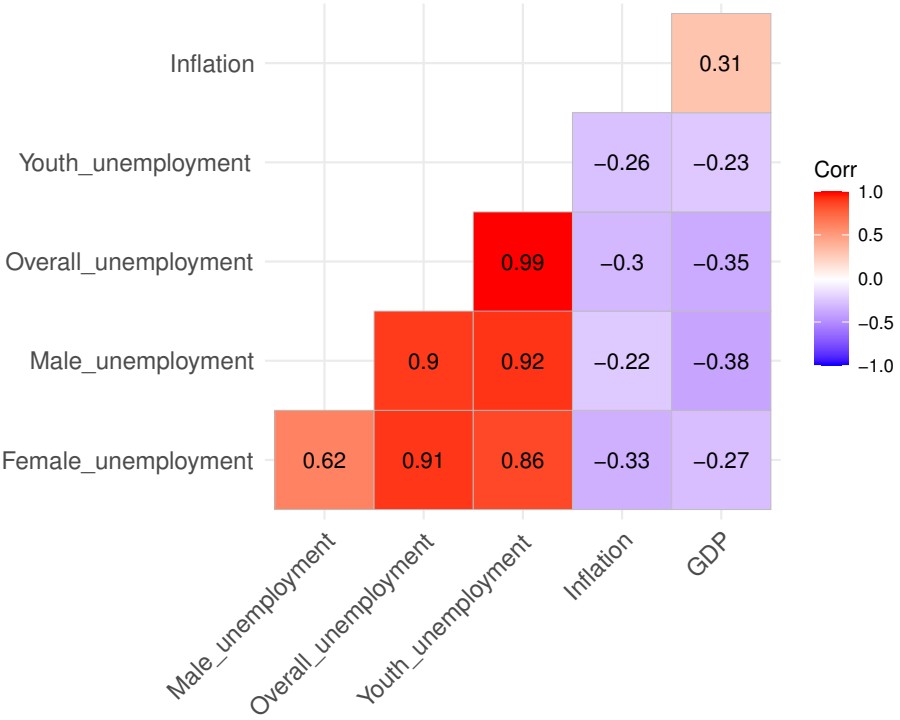

**Figure 1.** Heat map of the correlation.

Table 2 summarizes the findings of the unit root tests. The table shows the p-values for the Augmented Dickey-Fuller (ADF) test, Phillips-Perron (PP) test, Kwiatkowski-Phillips-Schmidt-Shin (KPSS) test, and Zivot-Andrews test (Dickey and Fuller 1979; Kwiatkowski et al. 1992; Phillips and Perron 1988). The outcomes of each unit root test show that, aside from inflation, which is stationary without differences, there are no unit roots in the differenced series. Traditional unit root tests, like the ADF, PP, and KPSS tests, do not consider structural breaks in the data. According to (Phillips and Perron 1988; Shiferaw 2023), applying these traditional unit root tests to data that contains structural breaks may produce misleading results. This evidence highlights the importance of taking into account unit root tests for structural breaks, such as the Zivot-Andrews test (Zivot and Andrews 1992), which allows for a single structural break in either the original or the differenced series. For instance, based on this test, the anticipated inflation break year is 2014.

**Table 2.** Unit root and stationary tests for the raw and differenced time series data.

| | **Original Time Series Data** | | | | **Differenced Time Series Data** | | | |
|---|---|---|---|---|---|---|---|---|
| | **ADF** | **PP** | **KPSS** | **Zivot-Andrews (Break Year)** | **ADF** | **PP** | **KPSS** | **Zivot-Andrews (Break Year)** |
| Total unemp | >0.1 | >0.1 | <0.05 | >0.1 (2018) | <0.01 | <0.01 | >0.1 | <0.01 (2005) |
| Youth unemp | >0.1 | >0.1 | <0.05 | >0.1 (2018) | <0.01 | <0.01 | >0.1 | <0.01 (2005) |
| Female unemp | >0.1 | >0.1 | <0.05 | >0.1 (2018) | <0.01 | <0.01 | >0.1 | <0.01 (2005) |
| Male unemp | >0.1 | >0.1 | <0.05 | >0.1 (2019) | <0.01 | <0.01 | >0.1 | <0.01 (2018) |
| Inflation | <0.01 | <0.01 | >0.1 | <0.01 (2014) | | | | |
| GDP | >0.1 | >0.1 | <0.05 | >0.1 (2003) | <0.01 | <0.01 | >0.1 | <0.01 (2005) |

### 3.1. Causality Tests

Before estimating a cross wavelet transform, the multivariate Student-$t$ GAS and ARDL models, the Toda–Yamamoto causality test (Toda and Yamamoto 1995) was performed to ascertain the relationship between the three indicators. This test has the benefit of being able to account for smooth transitional structural breaks. They developed a technique based on estimating the augmented structural vector autoregression (VAR) model $(k + d_{max})$, where $k$ is the optimal time lag on the first VAR model and $d_{max}$ is the maximum integrated order on the system's variables. The method examines the long-term relationships between the three indicator variables using annual data from 1991 to 2021. Table 3 displays the findings of the Toda–Yamamoto causality test over the three indicators (i.e., unemployment, inflation, and GDP). The results show that GDP is influenced by unemployment (i.e., overall employment, youth employment, and female employment). However, there is no Granger causal relationship between GDP and inflation, nor between unemployment and inflation.

**Table 3.** The Toda–Yamamoto causality test of unemployment, inflation and GDP.

| | **GDP** | **Inflation** | **Overall Unemp** | **Youth Unemp** | **Male Unemp** | **Female Unemp** |
|---|---|---|---|---|---|---|
| GDP | | 6.4 (0.38) | 3.5 (0.74) | 3.4 (0.76) | 3.7 (0.72) | 2.7 (0.84) |
| Inflation | 2.5 (0.87) | | 4.9 (0.55) | 6.4 (0.38) | 7.7 (0.26) | 4.1 (0.66) |
| Overall unemp | 16.0 * (0.014) | 1.4 (0.97) | | 7.8 (0.26) | 6.0 (0.42) | 6.5 (0.37) |
| Youth unemp | 14.7 * (0.022) | 0.54 (1.00) | 6.3 (0.39) | | 5.0 (0.54) | 5.2 (0.51) |
| Male unemp | 11.2 (0.082) | 4.9 (0.55) | 2.2 (0.91) | 2.1 (0.91) | | 2.3 (0.90) |
| Female unemp | 17.8 * (0.007) | 3.2 (0.78) | 8.9 (0.18) | 8.4 (0.21) | 8.4 (0.21) | |

Note that * denotes statistical significance at the 5% level of significance.

### 3.2. The Cross Wavelet Transform Analysis

Figure 2 shows the XWT findings for unemployment and inflation rates. The relationship between the inflation rate and the unemployment rate was out of phase from 1991 to

2003 and from 2005 to 2020, with a phase in 2020–2021. The four panels show an inverse relationship between unemployment and inflation rates. The XWT of unemployment rates and GDP log returns is shown in Figure 3. From 2003 to 2020, there was a clear lead/lag relationship between the unemployment rates and GDP log returns. More specifically, the XWT between GDP and the overall unemployment rate reveals that the two series move together positively from 1993 to 1996 and negatively from 2000 to 2003. Furthermore, the panels show an inverse relationship between unemployment and GDP from 2003 to 2020. The XWT between inflation and GDP log returns is shown in Figure 4. We can see from this graph that there was a clear lead/lag relationship between inflation and GDP log returns between 1995 and 2007 and 2012 and 2020. Specifically, the two series moved together upward for brief periods, such as 1993–1995 and 2010–2012. There have also been times when the two series moved in opposite directions, such as in 1991–1993, 2008–2010, and 2012–2013.

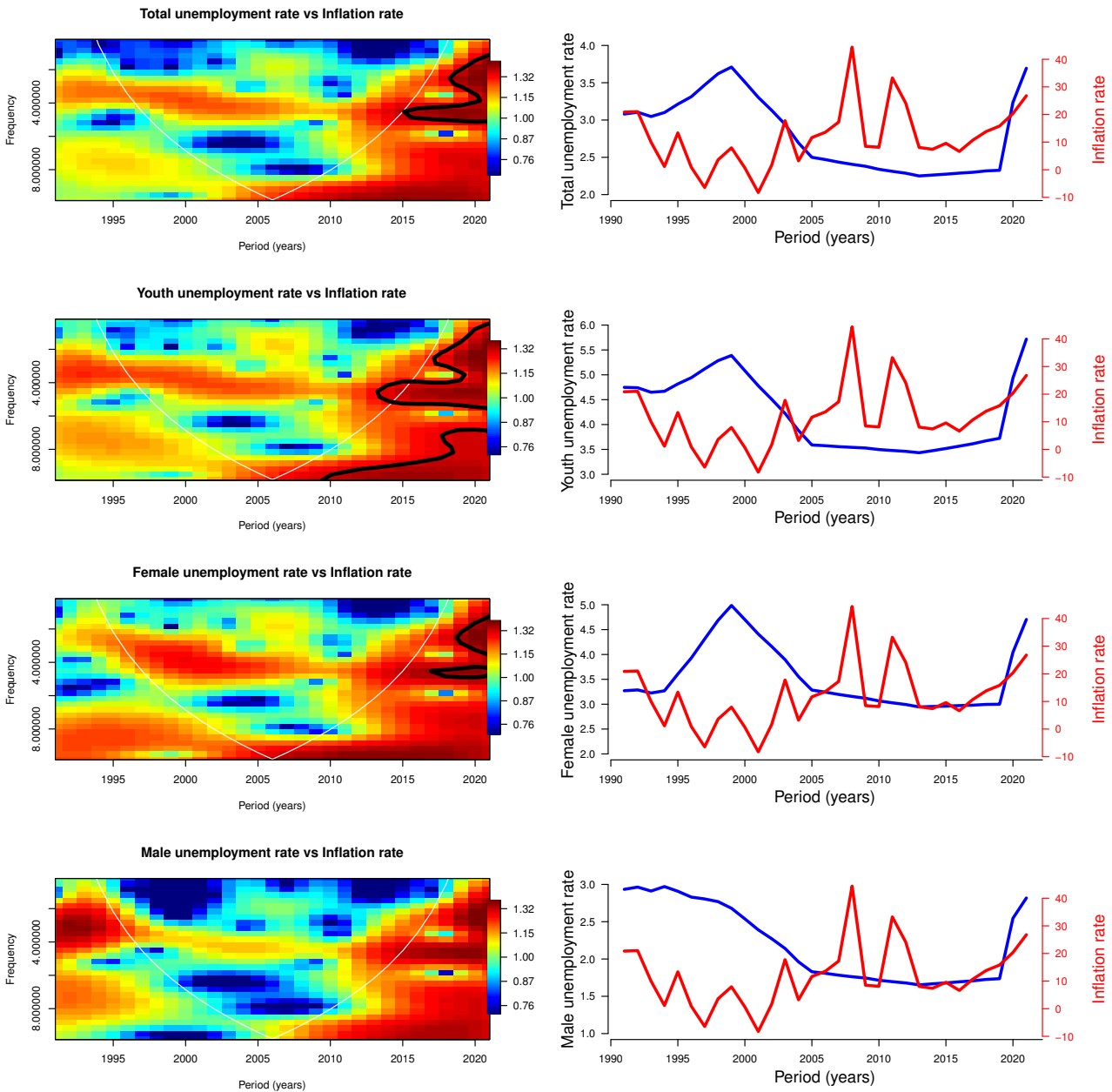

**Figure 2.** Cross-wavelet transform of unemployment and inflation.

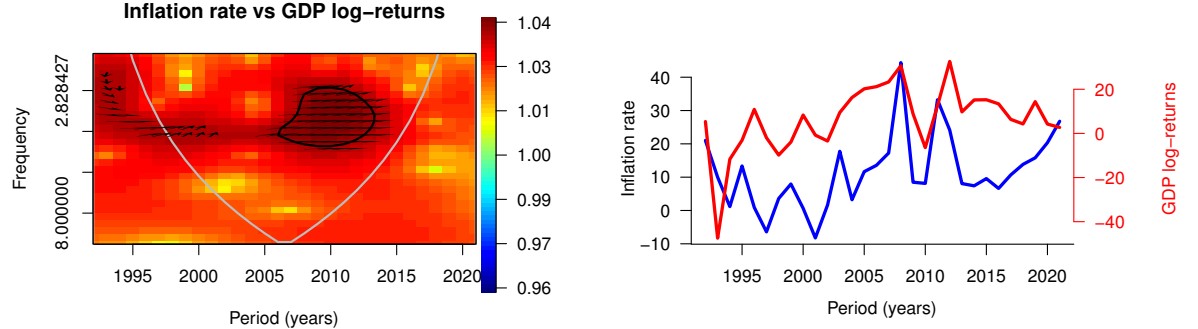

**Figure 3.** Cross-wavelet transform of unemployment and GDP log-returns.

**Figure 4.** Cross-wavelet transform of inflation and GDP log-returns.

### 3.3. The Multivariate GAS Model Results

Table 4 displays the parameter estimates from the multivariate GAS with Student-*t* model. The multivariate GAS(1,1) with Student-*t* model was found to be superior when evaluating the goodness-of-fit of this model using the Loglik value, the AIC, and the Schwartz Bayes Information Criterion (BIC). Additionally, the conditional correlation parameters a and b are significant, except for GDP and female unemployment rate and GDP and youth unemployment rate, indicating that the parameters change over time.

**Table 4.** The multivariate GAS with Student-*t* model estimation results.

| | GDP | | Unemployment | | | | | | | |
| | | | Overall | | Male | | Female | | Youth | |
| | a | b | a | b | a | b | a | b | a | b |
| Inflation | 0.3181 * | 0.8759 ** | 0.0001 ** | 0.6088 ** | 0.0001 ** | 0.4082 ** | 0.0001 ** | 0.7104 ** | 0.0422 ** | 0.6173 ** |
| GDP | | | 0.0175 * | 0.8690 ** | 0.1239 ** | 0.9081 ** | 0.0344 | 0.9138 ** | 0.0346 | 0.9634 ** |

\* Significant at 0.05 and ** significant at 0.01.

### 3.4. ARDL Results

The ARDL(3,3,3) model performed the best based on the AIC values. The ARDL(3,3,3) model explicitly fits the data the best because adding lags of one year, two years, and three years as explanatory variables improves the fitness of the yearly data. In the ARDL model analysis, the severity of multicollinearity was determined using the variance inflation factor (VIF). The first, second, and third lagged periods of unemployment all have VIF values greater than 5, which suggests the possibility of multicollinearity. Thus, excluding them solves multicollinearity by choosing the appropriate lag length of variables. Table 5 displays the outcomes of the fitted ARDL model. According to the DW values in Table 5, the estimation error residuals of the fitted ARDL model did not exhibit any serial correlation. The current unemployment rate had a significant negative impact on the current GDP (i.e., $GDP_t$). However, the current inflation rate had a positive impact on the current GDP (i.e., $GDP_t$). The inflation two years before (L(Infl,2)) had a significant negative impact on the current GDP (i.e., $GDP_t$). The value of DW is 2.03, which is near two, indicating that there is no autocorrelation. The adjusted R-squared value is at a moderate level of 64.30%, and the model is significant at the 5% level (*p*-value = 0.0003).

**Table 5.** The ARDL(3,3,3) model.

| | Estimate | Std. Error | t Value | *p*-Value |
| --- | --- | --- | --- | --- |
| Intercept | 42.8412 | 11.7861 | 3.6350 | 0.0018 ** |
| L(GDPt, 1) | 0.2636 | 0.2021 | 1.3040 | 0.2085 |
| L(GDPt, 2) | −0.0527 | 0.1134 | −0.4650 | 0.6477 |
| L(GDPt, 3) | −0.2297 | 0.1155 | −1.9890 | 0.0621 |
| Unempl | −5.9593 | 1.6660 | −3.5770 | 0.0021 ** |
| Infl | 0.4657 | 0.1455 | 3.2010 | 0.0049 ** |
| L(Infl, 1) | 0.1015 | 0.1562 | 0.6490 | 0.5242 |
| L(Infl, 2) | −0.3496 | 0.1485 | −2.3540 | 0.0302 * |
| L(Infl, 3) | −0.0897 | 0.1488 | −0.6030 | 0.5540 |
| Adjusted R-squared | 0.643 | | | |
| F-statistic | 6.854, | | *p*-value = 0.0003 | |
| DW | 2.03 | | | |

Note that ** and * denote statistical significance at the 1% and 5% levels, respectively.

## 4. Discussion

This study uses data from three decades on three leading indicators to examine the relationship between Ethiopia's unemployment, GDP, and inflation. The correlation parameters *a* and *b* between unemployment and inflation are highly significant (Table 4), implying that the correlation is dynamic (changing over time) (Meloni et al. 2022). Furthermore, as shown in Figure 1, inflation and unemployment have an inverse relationship. This is because more sectors hesitate to act thoroughly on incorporating inflation expectations into nominal wage demands when they experience unemployment (Palley 2012). Simona et al. (2019) also argued that the inflation rate hurts foreign direct investment (FDI), increasing unemployment because FDI flows are widely believed to create new jobs and thus reduce unemployment.

In the unemployment-GDP pair, the correlation parameters a and b (see Table 4) are highly significant, indicating that the correlation is dynamic. Additionally, for every 1% increase in unemployment, GDP growth decreased by roughly 6% in the fitted ARDL (3,3,3) model, controlling for changes in inflation and its lags, the first, second, and third lags of GDP. This is unsurprising given the effects of GDP growth on unemployment (both short- and long-term) (Meloni et al. 2022). Several empirical studies have investigated the relationship between unemployment and GDP. For example, Okun (1962) studied the relationship between unemployment and GDP using the US dataset after the Second World War and found a negative relationship between the two indicators. Most recently, Bein and Ciftcioglu (2017) investigated the relationship between the relative share of agriculture in GDP and the unemployment rate for selected European countries and found that the unemployment rate is negatively related to the relative GDP share of agriculture.

The GDP-inflation pair's correlation parameters a and b (see Table 4) are highly significant, indicating a dynamic relationship. Additionally, controlling for changes in unemployment and the first, second, and third lags of GDP, the fitted ARDL(3,3,3) model showed that for every 1% increase in inflation, GDP growth increased roughly 0.4657%. However, this model demonstrates that the second inflation lag (two years prior) significantly negatively affected the current GDP, holding other explanatory variables constant. The most recent study, conducted by (Batrancea et al. 2022), used an econometric approach to examine the relationship between GDP growth and inflation and other explanatory variables. Based on their econometric analysis of inflation targeting policy in the emerging economies of Europe and Central Asia, (Arsić et al. 2022) found that inflation targeting has effectively reduced inflation, inflation volatility, and GDP volatility.

The unemployment-inflation pair exhibits a strong dynamic link from 1991 to 2012 (Figure 2); however, the relationship's occurrence is not immediately apparent. For a considerable time, the unemployment-GDP relationship has been strongly correlated. Notably, the time series' early eras showed significant interaction (i.e., during 1991–2000). Interestingly, in the early decades of the time series, the unemployment-inflation (Figure 2), unemployment-GDP (Figure 3), and inflation-GDP (Figure 4) pairs exhibit comparable patterns and a solid dynamic relationship. Specifically, the interaction was solid between 1991 and 2000. Ethiopia's considerable development obstacles can explain this during and after the military regime. By almost every primary indicator of economic growth during the seventeen years (1974–1991) of the centralized state of the military government (the Derg regime), the nation was in a terrible situation because of the war, environmental degradation, bad policies, uncontrolled population growth, and many other factors (Chole 2004).

In other words, Ethiopia's economy was in ruins after seventeen years of terrible conflicts under the Derg dictatorship (Milas and Latif 2000). With the help of donors, the transitional government established an emergency recovery program with a microeconomic reform program following the overthrow of the Derg regime in May 1991 and the victory of the EPRDF (Milas and Latif 2000). In September 1992, the new administration agreed to a policy framework document with the World Bank and the International Monetary Fund (IMF), which resulted in regulations and structural reforms from 1993 to 1996 (ADB 2000).

The economy did well between 1994 and 2004, with average annual increases in the total real GDP and the GDP per capita of 5% and 2%, respectively, (Bezu et al. 2012).

Given the initial findings, it is appropriate to present the following argument. The results of the multivariate GAS model (Table 4) show a significant relationship between the three indicators. The significant time-varying correlations between the three leading indicators can be explained by several factors, including a decline in agriculture production, changes in climate, public infrastructure investment policies, political unrest, the COVID-19 pandemic, a decline in employment in the manufacturing sector, and more.

### 4.1. A Decline in Agriculture Production

The mainstay of Ethiopia's economy has been agriculture (Wendimu 2021). Agriculture contributed 70% of the GDP in the 1960s, but that percentage fell to 52.3% in 1973–1974 and 44.3% in 2011–2012, respectively, (Mekuria 2018). In the recent past, agriculture provided a living for roughly 85% of the population, over 40% of the GDP, and around 60% of foreign exchange profits (International Energy Agency 2019; Makombe et al. 2007). From 1984 to 2007, the GDP and agriculture reliance ratios were 49.8% and 45.0%, respectively, although the labor force has expanded from 50.2% to 51.9% (Rodrik 2018). However, due to irregular rainfall patterns, soil erosion, insufficient irrigation, land degradation, and insufficient farm inputs, agriculture showed a decreasing trend, which should result in a slowdown in economic growth and rising unemployment (Mekuria 2018).

### 4.2. Climate Change

As mentioned in the preceding paragraph, Ethiopia heavily depends on agriculture for employment. However, because approximately 96% of the cultivated land depends on rainfall, Ethiopia is more at risk from climate change (Khatami et al. 2020; Wendimu 2021). The country will be especially susceptible to the effects of climate change, which will impact the agriculture, energy, and health sectors through droughts, flooding, desertification, water scarcity, and increased pest activity (ADB 2022). For instance, the 2016 El Nino climatic cycle variation resulted in drought. About 10.2 million people were affected by this drought, which necessitated a $1.9 billion humanitarian response (ADB 2022). A rapidly growing population, rising poverty, and an unstable economy will also impact the nation.

### 4.3. Political Instability

The economy shrank and foreign exchange reserves dropped due to the civil war between the EPRDF and TPLF, putting the new regime's efforts to expand the economy and create jobs in jeopardy (Meester et al. 2022). All sides' combatants killed civilians, including children, and purposefully obstructed aid deliveries. Ethiopia, a nation that has long been regarded as a cornerstone for EU relations with the Horn of Africa, no longer receives the majority of the budget support from the EU (Pichon 2022).

### 4.4. The COVID-19 Pandemic

Ethiopia's economic performance has suffered due to COVID-19, and the country has experienced a drop in external demand since April 2020. In early 2020, merchandise exports, excluding gold, rose by 5.8%, but between July and December, they fell by 4.1%. As a result of the outbreak, clothing, textiles, and fruits and vegetables have taken a brutal hit. The service sector experienced negative growth in exports and imports in 2020, while remittances fell by 10% (Sánchez-Martín et al. 2020). In general, COVID-19's effects on transportation and hospitality caused the country's economy to slow from 6.1% growth in 2020 to 5.6% growth in 2021 (ADB 2022).

### 4.5. Public Infrastructure Investment Strategy Concerns

Public infrastructure investment, which increased from 5% to 19% of GDP, has been the main driver of Ethiopia's astonishing GDP growth since 2004 (International Energy Agency 2019). Additionally, the 45% shift in the GDP's composition from agriculture to

services led to moderate transfers in labor from agriculture to construction and services (Rodrik 2018). Consequently, there are worries regarding the strategy's viability, given that public investments are only sometimes successful (Rodrik 2018).

*4.6. The Manufacturing Sector Accommodates Less Labour Force*

The number of businesses, sales, and jobs in Ethiopia's manufacturing sector have all increased significantly over the past 20 years. However, the worst news is that only about 6% of formal employment in Ethiopia is in the manufacturing sector (Rodrik 2018). Even by African standards, it continues to be a relatively small sector in terms of contributions to GDP and employment, and it has yet to start to become export-oriented. Africa's unfavorable investment climate is frequently blamed partly for the continent's industrial sectors' slow growth (Shiferaw and Söderbom 2018).

## 5. Conclusions and Policy Implications

This article aims to study the dynamic interactions among Ethiopia's leading indicators, such as unemployment, inflation, and GDP, that the general public and policymakers pay special attention to and monitor by analyzing the annual time-series data spanning 1991 through 2021. These indicators give an economy a score and a general sense of its performance (Debaere 2008). The study used the TY, WSC, multivariate GAS with Student-*t*, and ARDL approaches to analyze the dynamic relationship between the three economic pillars.

The TY method reveals a correlation between GDP and all types of unemployment, including youth unemployment, female unemployment, and unemployment overall. In other words, unemployment (overall, youth and female) Granger causes GDP. GDP and inflation, as well as unemployment and inflation, have a neutral relationship. Additionally, the ARDL method found that while inflation had a significant positive impact on GDP, overall unemployment had a significant negative impact.

For multivariate Student-*t* GAS model results, the key findings from this model indicate that the correlation between unemployment (overall, male, female, and youth) and inflation is highly significant. Except for female and youth unemployment, there is a significant correlation between unemployment and GDP. Additionally, a significant correlation between GDP and inflation suggests that this relationship changes over time (i.e., dynamic).

According to the results of the XWT analysis, the unemployment-inflation pair showed relatively strong comovement in the XWT results after 2005. In contrast, the unemployment-GDP pair moved in the other direction for roughly the whole sample period. Inflation and GDP, however, exhibit a high comovement in the early part of the sample (over the period 1991–2000) but a poor comovement in the later parts of the sample (during the period 2012–2021). The comovement strength generally varies depending on the frequency bands and the inflation, GDP, and unemployment periods.

Due to several factors, including its detrimental effects on people's financial well-being and the national budget, unemployment is a severe social and economic problem in most developing countries, including Ethiopia. This demonstrates why achieving full employment is a macroeconomic goal, making unemployment one of the most critical indicators of an economy's health (Ogbeide et al. 2016). Approximately 85% of Ethiopia's population depends on agriculture for their livelihood. The sad fact is that Ethiopia's agriculture is primarily dependent on rainfall. The nation's susceptibility to climate change has been and will be present in the future. To resolve this dilemma, the Ethiopian government should adopt programs that are driven by agriculture and are climate change-adaptive (e.g., irrigation development). Additionally, the nation has abundant water resources that may be exploited for agriculture, ironically dubbed the "water tower" of Eastern Africa.

Ethiopians recently had the most exciting news about the two-year civil war between the federal government and the TPLF. After talks in South Africa between the warring groups in Ethiopia, the two-year conflict that may have cost hundreds of thousands of lives

officially ended on 2 November 2022. Ethiopians anticipate that this agreement will help restore the nation's peace and security and the most severely damaged economy. Mekelle, the capital of the Tigray region, was most recently made accessible to the federal government.

Further research should examine the interactions between climate change, agriculture, and unemployment in Ethiopia. The main limitation of this paper is the need for high-frequency time series data. This paper used the most recent data (1991–2021) to examine the relationship between unemployment, inflation, and GDP in Ethiopia. Despite this limitation, this study is one of the first to illustrate the dynamic relationship between Ethiopia's three pillars of the economy (i.e., inflation, GDP and unemployment).

**Funding:** This research received no external funding.

**Informed Consent Statement:** Not applicable.

**Data Availability Statement:** The data presented in this study are available on request from the corresponding author.

**Conflicts of Interest:** The author declares to have no competing interests.

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
