# Peer review of "An Understanding of How GDP, Unemployment and Inflation Interact and Change across Time and Frequency"

_economies, doi:10.3390/economies11050131_

Round 1

Reviewer 1 Report

1. Abstract should be streamlined to show Context and Scenario.

2. The Problem must be inserted into the Abstract.

3. Introduction did not show the negative externalities of the issues. The author needs to streamline the flow to lead to the why of the Article. It is not clear at the moment.

4. Literature must indicate the Ambivalences from previous studies.

5. A brief justification why the chosen methodology is required, because there are other ways to do the analysis.

6. In the Discussion, a little comparative analysis is needed as per present findings and previous findings. This will situate the Manuscript properly and its contribution to knowledge.

Reviewer 2 Report

Dear authors, 

Please find my comments below:

General Comments

From my point of view, it is a very interesting topic and simultaneously it seems that to the best of my knowledge is the first empirical study to applies the cross-wavelet transform (XWT) technique and multivariate Student-t Generalized Autoregressive Score (GAS) model to investigate the dynamics of Ethiopia’s three main economic pillars: unemployment, inflation, and GDP. The findings from the XWT approach demonstrate a relationship between unemployment, inflation, and GDP that is dynamic and varies over time and frequency. Additionally, the multivariate Student-t GAS model results demonstrate a significant time-varying correlation among the three indicators, demonstrating a solid and dynamic relationship between them. Overall, the study’s findings point to a considerable negative impact of unemployment on both GDP and inflation.

1.      The abstract must contain the main purpose of the paper, the research method used in the research and the main contributions.

2.      It would be very useful to add in the "Introduction" section the purpose, objectives and hypothesis of the research. I consider that a weak point of the paper is that the authors did not show the novelty of the paper compared to other works. The introduction should specify the novelty of the paper compared to other papers published in this area.

3.      The research is well based on science and the results are in agreement with the theoretical part. From my point of view, the paper is original and the topic addressed brings added value to the specialized literature regarding the influences of  unemployment, inflation on economic growth. The paper is well written and easy to read.

4.      At the same time, the authors are required to present Descriptive Statistics, Correlation matrix with all tests and indicators: standard deviation, Skewness and Kurtosis interpretation, Jarque-Berra with probabilities, etc.

5.      It is necessary for the authors to do a panel-type econometric analysis in order to increase the quality of the paper considering the analyzed period 1991-2021.

6.      It is important to present the VIF test on multicollinearity between independent variables. Heteroskedasticity and Endogeneity tests are also important in this study. All these aspects that are not found in the paper represent weaknesses of the research.

7.      At the same time, I consider that the conclusions part of the work should be expanded.

8.      I think that the literature needs to be improved with other works, refered to economic growth.  That is why I recommend the authors to refer to other recent works indexed in Web of Science. I suggest that the authors cite papers published in Web of Science Journals, such as:

Batrancea, L., Rathnaswamy, M.K. & Batrancea, I. A Panel Data Analysis on Determinants of Economic Growth in Seven Non-BCBS Countries. J Knowl Econ (2021). https://doi.org/10.1007/s13132-021-00785-y

Batrancea L.M. (2021) An Econometric Approach on Performance, Assets, and Liabilities in a Sample of Banks from Europe, Israel, United States of America, and Canada. Mathematics, 9(24):3178. https://doi.org/10.3390/math9243178.

Batrancea, L. (2021) The Influence of Liquidity and Solvency on Performance within the Healthcare Industry: Evidence from Publicly Listed Companies,  Mathematics 9, no. 18: 2231. https://doi.org/10.3390/math9182231, eISSN:2227-7390/sept.2021

9.      The conclusions at the end of the paper should be expanded showing the economic policy implications of the research results.

In conclusion, the article should be improved. It should also be enhanced with a review of the literature adequate to the subject and a broader interpretation and commentary of the research results.

Reviewer 3 Report

Here are some suggestions and comments to further improve:

1) The issue is interesting. The Ethiopian case can be interesting for other developing nations of the region. 

2) The title suggests the paper is driven by methods rather than the issue.   

3) The motivation of the paper is not clearly explained as there are many similar studies. Please clearly highlight the contribution of the paper. How your work is different from others? Contribution could be theoretical or empirical. 

4)   In many places author(s) mentioned the word impact. However, correlation does not imply causation, as there could be other factors or variables influencing the relationship between the two variables. Establishing causality requires more evidence than just observing a correlation, such as experimental manipulation or controlling for other possible confounding variables.

5) Although a sample size of at least 30 is recommended for reliable results in time series analysis but the use of annual data in XWT to investigate time and frequency domain remain confusing. 

6) In figure 2, the quality of images (left panels) are very poor. 

Round 2

Reviewer 3 Report

The author(s) have made significant improvements in the manuscript. Interestingly, the author(s) add ARDL results as a robustness check. However, it is important to check and present the stationarity of the variables in level and differenced form. As it is well established in the literature that many financial variables are non-stationary in the level form but become stationary in the first differenced form which is an important prerequisite to run ARDL.

Besides this, I don't have any other comments or suggestions.
